# Fully Automated Production of [^68^Ga]GaFAPI-46 with Gallium-68 from Cyclotron Using Liquid Targets

**DOI:** 10.3390/ijms242015101

**Published:** 2023-10-12

**Authors:** Alexandra I. Fonseca, Vítor H. Alves, Ivanna Hrynchak, Francisco Alves, Antero J. Abrunhosa

**Affiliations:** 1ICNAS Pharma, University of Coimbra, 3004-531 Coimbra, Portugal; alexandrafonseca@icnas.uc.pt (A.I.F.); vitoralves@uc.pt (V.H.A.); ivanna.ua@icnas.uc.pt (I.H.); 2Fluidomica, Lda., 3060-197 Cantanhede, Portugal; 3CIBIT/ICNAS, Institute for Nuclear Sciences Applied to Health, University of Coimbra, 3004-531 Coimbra, Portugal; franciscoalves@uc.pt; 4Instituto Politécnico de Coimbra, ESTeSC—Coimbra Health School, 3045-093 Coimbra, Portugal

**Keywords:** radiopharmaceuticals, cyclotron, liquid targets, GaFAPI-46

## Abstract

^68^Ga-based radiopharmaceuticals are routinely used for PET imaging of multiple types of tumors. Gallium-68 is commonly obtained from ^68^Ge/^68^Ga generators, which are limited in the quantity of activity produced. Alternatively, gallium-68 can easily be produced on a cyclotron using liquid targets. In this study, we optimized the GMP production of [^68^Ga]GaFAPI-46 using gallium-68 produced via a standard medical cyclotron using liquid targets. Starting from the published synthesis and quality control procedures described for other ^68^Ga-based radiopharmaceuticals, we have validated the synthesis process and the analytical methods to test the quality parameters of the final product to be used for routine clinical studies. [^68^Ga]GaFAPI-46 was successfully produced with high radiochemical purity and yield using an IBA Synthera^®^ Extension module. Gallium chloride was produced on a medical cyclotron using a liquid target with activity of 4.31 ± 0.36 GBq at the end of purification (EOP). Analytical methods were established and validated, meeting Ph. Eur. standards. Full GMP production was also validated in three consecutive batches, producing 2.50 ± 0.46 GBq of [^68^Ga]GaFAPI-46 at the end of synthesis (EOS), with 98.94 ± 0.72% radiochemical purity measured via radio-HPLC. Quality was maintained for up to 3 h after the EOS. Production of [^68^Ga]GaFAPI-46 was performed and validated using a standard medical cyclotron with liquid targets. The quality control parameters (e.g., sterility, purity, and residual solvents) conformed to Ph. Eur. and a shelf life of 3 h was established. The activity of [^68^Ga]GaFAPI-46 produced was substantially higher than the one obtained with generators, enabling a better response to the clinical need for this radiopharmaceutical.

## 1. Introduction

Cancer-associated fibroblasts (CAFs) play an important role in the tumor microenvironment, exhibiting functions in tumor migration, growth, metastasis, progression, and resistance to chemotherapy [1]. Fibroblast activation protein (FAP) is a membrane-bound glycoprotein that is specifically overexpressed in activated fibroblast, including CAFs [2]. FAPI-46, in particular, has emerged as a very promising theranostic tool in cancer. When labeled with the positron-emitting gallium-68, FAPI-46 has showed improved tumor-to-organ ratios, allowing for high-contrast PET imaging of multiple types of tumors [3]. In this context, there have been several attempts to optimize the GMP production and quality control of [^68^Ga]GaFAPI-46, and its therapy counterpart [^177^Lu]LuFAPI-46, to meet the increasing clinical demand for these radiopharmaceuticals [4,5,6,7].

Gallium-68 is available worldwide through the ^68^Ge/^68^Ga generator [8]. Despite being convenient, generators have limitations in terms of the maximum number of elutions and activity per elution, high cost, and the possibility of contamination with long-lived parent radionuclide germanium-68 (half-life 271 days) [9]. In contrast, cyclotron production of gallium-68 takes advantage of the extensive network of medical cyclotrons available that can produce considerable amounts and perform consecutive production cycles [10]. To our knowledge, the production of [^68^Ga]GaFAPI-46 using gallium-68 from a cyclotron has not been reported. The validation of its GMP production can help us to meet the growing demand for this radiopharmaceutical worldwide. In this work, we describe a method for the GMP-automated synthesis and full validation of this process. We believe that this could serve as a roadmap for other laboratories that are trying to implement the routine synthesis of this radiopharmaceutical.

## 2. Results

### 2.1. [^68^Ga]GaFAPI-46 Synthesis

Gallium-68 was produced using a standard medical cyclotron (IBA Cyclone Kiube, Louvain-la-Neuve, Belgium) via the irradiation of a zinc-68 solution for 70 to 80 min. Purification and synthesis were performed on an IBA Synthera^®^ Extension fully automated platform (Louvain-la-Neuve, Belgium). The starting activity at the EOP was 4.31 ± 0.36 GBq (*n* = 3), and the peptide quantity was 50 µg/batch (11.67 ± 0.97 µg/GBq). All syntheses were successfully completed within 25 min after the EOP, and all quality control parameters were in line with Ph. Eur. specifications. Notably, the highest activities in the final product vial, with approximately 2.5 GBq of [^68^Ga]GaFAPi-46, produced comparable results in terms of radiochemical purity to the lower activity vials (Table 1). Table 2 summarizes the results obtained from the Spreckelmeyer (2020) and Alfeimi (2022) studies, which used different synthesis modules and generator-produced gallium-68 to synthesize [^68^Ga]GaFAPI-46 [6,7]. In comparison to these studies, we achieved similar results in terms of decay-corrected radiochemical yield (RCY) and radiochemical purity (RCP), with values of 90.53% and 98.94% using radio-High-Performance Liquid Chromatography (radio-HPLC) respectively, but with up to three times more activity at the EOS.

### 2.2. Validation of Analytical Methods

Validation was performed for the three key QC techniques used to assess the quality of [^68^Ga]GaFAPI-46: HPLC, Thin-Layer chromatography (TLC), and Gas chromatography (GC). Method validation and analysis for sterility were conducted by a certified outsourcing company (MicroBios, Barcelona, Spain). pH and endotoxin analysis were performed as described in the relevant monographs of Ph. Eur. [11]. The HEPES test was based on previously established procedures for other ^68^Ga-based radiopharmaceuticals (e.g., monograph of GALLIUM (^68^Ga) PSMA-11 INJECTION).

#### HPLC, TLC, and GC Methods Validation

HPLC was used to identify and quantify radiochemical impurities in the drug product (Figure 1). The radio-TLC technique was used to establish the radiochemical purity of [^68^Ga]GaFAPI-46 (Figure 2), and GC to quantify the presence of ethanol in the final formulation. To validate the analytical methods, the following parameters were checked: accuracy, repeatability, selectivity/specificity, quantification limit (LOQ), linearity, and range [12]. A summary of the validation of HPLC, TLC, and GC methods results can be seen in Table 3.

### 2.3. Quality Control

Quality control (QC) tests for [^68^Ga]GaFAPI-46 were established based on the current requirements for other ^68^Ga-based radiopharmaceuticals monographs (e.g., monograph of GALLIUM (^68^Ga) PSMA-11 INJECTION). Table 4 displays the results of three exemplificative QC batches of [^68^Ga]GaFAPI-46, which were also used to validate the synthesis process of the final drug product.

### 2.4. [^68^Ga]GaFAPI-46 Stability

The stability of [^68^Ga]GaFAPI-46 in 10% (*v*/*v*) ethanol formulation at room temperature was tested for up to 3 h using radio-HPLC; [^68^Ga]GaFAPI-46 was not stable under these conditions at higher activities. During this time period, two radioactive side-products were detectable, with increased percentages at higher activities. The stability of [^68^Ga]GaFAPI-46 was achieved by adding 500 mg of sodium ascorbate to the final formulation. The radiochemical purity of all batches remained above 95% stable over a 3 h period of incubation at room temperature (RT), regardless of vial activity (Figure 3).

## 3. Discussion

Typically, the synthesis of ^68^Ga-based radiopharmaceuticals is performed using 1.85 GBq (from the generator elution). Protocols for fully automated production of [^68^Ga]GaFAPI-46 using generators have been published with up to 1.7 GBq of [^68^Ga]GaFAPi-46 [7] and, generally, over 90% radiochemical yield (Table 2). In this article, we present the results and validation of a fully automated synthesis method of [^68^Ga]GaFAPI-46 produced from a cyclotron. Furthermore, we have validated the specific QC analytical methods for this tracer.

When using gallium-68 from a cyclotron to produce [^68^Ga]GaFAPi-46, the starting activities of gallium-68 can be as high as 5 GBq after purification. With the increasing demand for ^68^Ga-based radiopharmaceuticals in the last decade, cyclotron production of gallium-68 enables a better response to the clinical doses of gallium-68 which are routinely necessary. The automated synthesis of [^68^Ga]GaFAPI-46 using a Synthera^®^ Extension (Table 1) was found to be highly reproducible. Our results demonstrate that the use of 50 µg of FAPI-46 yields a similar radiochemical yield as in previously described studies [6,7], even with triple the activity. 

The addition of sodium ascorbate in the final formulation of the product prevented radiolysis of the radiopharmaceutical. The formulation of [^68^Ga]GaFAPI-46 in saline with 10% vol. ethanol was found to be unstable. Although this effect has already been formerly described, it had not previously been observed to this extent. The use of ascorbic acid in the reaction has been reported to improve stability for ^68^Ga- and ^177^Lu-based radiopharmaceuticals [4,5,13]. Additionally, the use of sodium ascorbate in the final formulation has been shown to enhance stability in [^177^Lu]LuFAPI-46. Consistent with these findings, the use of sodium ascorbate prevented degradation of [^68^Ga]GaFAPI-46 over a 3 h period, even at high activities of up to 3.0 GBq. Results of synthesis validation, summarized in Table 4, demonstrate that all the tested quality parameters were in accordance with the Phar. Eur.

## 4. Materials and Methods

The FAPI-46 precursor and the standard [^nat^Ga]Ga-FAPI-46 were manufactured by ABX (Radeberg, Germany) and were made available, free of charge, by SOFIE Biosciences (Dulles, VA, USA). An aqueous stock solution of 1 mg/mL was prepared, and aliquots of 50 µg were stored at −15 °C. All chemicals were of analytical grade, and the solvents for high-pressure liquid chromatography (HPLC) were purchased as HPLC grade. Enriched zinc-68 (66 mg/mL and 98.0% isotopic enrichment) for gallium-68 production, as well as all the chemicals and tubing kits for gallium-68 purification, were purchased from Fluidomica (Cantanhede, Portugal).

### 4.1. Irradiation and Purification of [^68^Ga]GaCl_3_

Irradiation of zinc-68 liquid targets, and further gallium-68 purification, was conducted following the previously published and described methodology [10,14]. Briefly, gallium-68 was obtained from the irradiation of enriched zinc-68 solutions using an IBA Cyclone Kiube (IBA—Louvain-la-Neuve, Belgium). Zinc-68 was supplied by Fluidomica (Cantanhede, Portugal) dissolved in 0.01 M nitric acid, yielding a concentration of 66 mg/mL. These solutions were irradiated with 13 MeV protons for 70 min. The resulting target solution was transferred to a shielded hot-cell, and gallium-68 automatic purification was conducted using a Synthera^®^ Extension module (IBA—Louvain-la-Neuve, Belgium) without any manual intervention. 

To proceed with the purification, the target solution was dissolved in water and loaded onto a cation exchange resin (SCX), which had been preconditioned with 10 mL of 3 M HCl followed by 10 mL of water. The resin was then washed with a mixture of acetone and HBr to remove any zinc traces. Finally, gallium-68 was eluted with 3 M HCl and mixed with 10 mL of concentrated HCl. This solution was then loaded onto a strong cation exchange (SAX) column, dried for 2 min with inert gas, and eluted with 10 mL of water in the final collection vial. All reagents and tubing kit were supplied by Fluidomica (Cantanhede, Portugal).

### 4.2. Synthesis of [^68^Ga]GaFAPI-46 Using a Synthera^®^ Extension Synthesizer

For the fully automated synthesis of [^68^Ga]GaFAPI-46 using an IBA (Louvain-la-Neuve, Belgium) Synthera^®^ Extension module, shown in Figure 4a, we used single-use labeling cassettes and reagent kits supplied by Fluidomica. Each reagent kit included a SXC bound elute cartridge, a C18 plus short cartridge, HEPES buffer (0.5 M), saline, ethanol, and a sodium ascorbate vial (500 mg). The C18 cartridge required preconditioning with 10 mL of ethanol and 10 mL of water and drying with air before use. Before gallium-68 purification, the reaction mixture, consisting of 1 mL of 0.5 M HEPES buffer, 10 mg of ascorbic acid, and 50 µg of FAPI-46, was introduced into the reactor vial. 

General automated synthesis/radiolabeling steps: The C18 plus short cartridge is preconditioned with ethanol (10 mL) followed by water (10 mL) prior to use.Next, 50 µg of FAPI-46 precursor, dissolved in 1 mL of 0.5 M HEPES, is added to the reaction vial.Purified gallium-68 (10 mL) is loaded into the SCX bound elute cartridge using a peristaltic pump to prevent cross-contamination of the tubing system.The loaded SCX cartridge is eluted with a 5 M NaCl (in 0.05 M HCl) solution in the reactor vial.Radiolabeling reaction takes 5 min at a 90 °C temperature.Reaction mixture is cooled down with water (5 mL) and passed through the C18 plus short cartridge, at 5 mL/min flow, to the waste container.C18 cartridge is then rinsed with water (10 mL) at a 5 mL/min flow.Finally, [^68^Ga]GaFAPI-46 is eluted from the C18 column with a solution of 2 mL water/EtOH (1:1) and filtered into the final product vial, which is infused with 500 mg of sodium ascorbate.After purification and synthesis, [^68^Ga]GaFAPI-46 is transferred to the Quality Control (QC) laboratory and all the components are measured, after which the decay-corrected RY is determined.

### 4.3. Radionuclidic Identity and Purity

#### 4.3.1. HPGe Analysis

The radionuclidic purity (RNP) of gallium-68 at the end of beam (EOB) was determined through γ-spectroscopy of the final solution, using a High-Purity Germanium detector (HPGe) several hours after the EOB. The HPGe was calibrated with ^154^Eu and ^133^Ba radioactive sources and subjected to low-background shielding. γ-spectra were acquired using point-source-like samples with a dead-time below 4%. GammaVision (ORTEC Inc., Easley, SC, USA) software (for Windows Model A66-B32 Version 6.07) was used to determine photopeak areas.

#### 4.3.2. Half-Life Measurements

The half-life was determined by measuring gallium-68 in the ionization chamber with a time interval of 5 min.

### 4.4. Radiochemical Purity and Identity

#### 4.4.1. HPLC Analysis

The HPLC method used in this study was previously described by Eryilmaz et al. [5]. Table 5 specify the equipment and operating conditions used during all HPLC analyses.

#### 4.4.2. TLC Analysis

As described in the European Pharmacopeia (Ph. Eur.), an ammonium acetate (77 g/L):methanol (50:50 *v*/*v*) solution was used as the mobile phases for iTLC, and iTLC-SG strips were used as the stationary phases. Table 6 specify the equipment and operating conditions used during all TLC analyses. The colloidal species of gallium-68 were detected at Rf < 0.1 and the product [^68^Ga]GaFAPI-46 was detected at Rf > 0.5.

### 4.5. Residual Solvents

#### 4.5.1. Ethanol

The presence of ethanol was evaluated via gas chromatography (GC) using an Agilent 6850 RaytestGmbh (Straubenhardt, Germany) GC system. Table 7 specify the equipment and operating conditions used during all GC analyses.

#### 4.5.2. HEPES

This system uses TLC aluminum foil as the stationary phase and a mixture of water and acetonitrile (25:75 *v*/*v*) as the mobile phase. A reference solution containing 200 µg of HEPES in 10 mL of water was eluted in the strip along with the sample and the strip is then exposed to iodine vapor for 4 min for HEPES detection.

### 4.6. Stability of [^68^Ga]GaFAPI-46

The stability of [^68^Ga]GaFAPI-46 in its final formulation, consisting of 10% EtOH (*v*/*v*) with ascorbic acid, was evaluated via HPLC, and the stability measurements were quantified using the previously validated method. 50 µL aliquots were taken at different time points and measured using the HPLC method up to three hours after the end of synthesis.

## 5. Conclusions

In this study, we demonstrated that [^68^Ga]GaFAPI-46 can be produced according to GMP using liquid targets in a medical cyclotron, resulting in significantly higher production levels compared to ^68^Ge/^68^Ga generators. The high radionuclidic and radiochemical purity, as well as the stability of the final drug product, indicate that this method could serve as an alternative to conventional gallium-68 generators. These findings provide a roadmap for future [^68^Ga]GaFAPI-46 implementations aimed at meeting the routinely necessary clinical dose requirements.

## Figures and Tables

**Figure 1 ijms-24-15101-f001:**
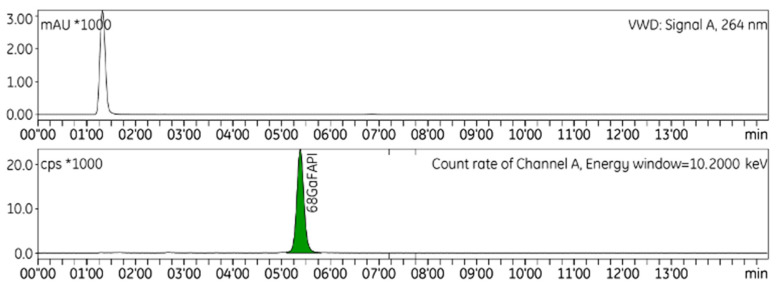
Representative radio-HPLC chromatogram of [^68^Ga]GaFAPI-46.

**Figure 2 ijms-24-15101-f002:**
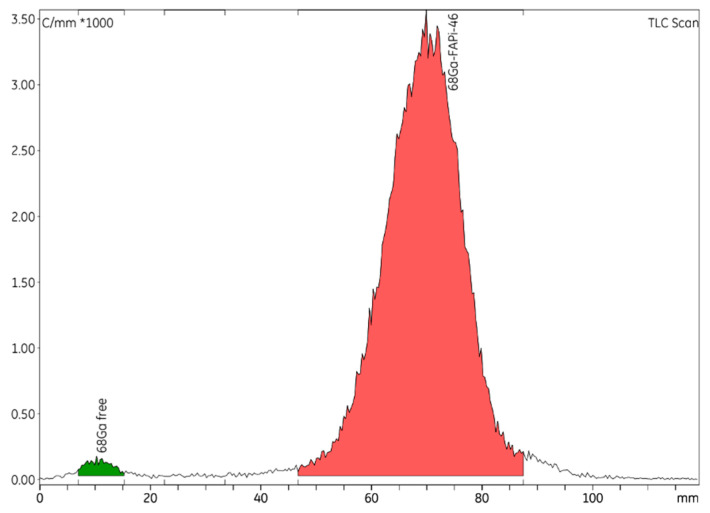
Representative radio-TLC chromatogram of [^68^Ga]GaFAPI-46.

**Figure 3 ijms-24-15101-f003:**
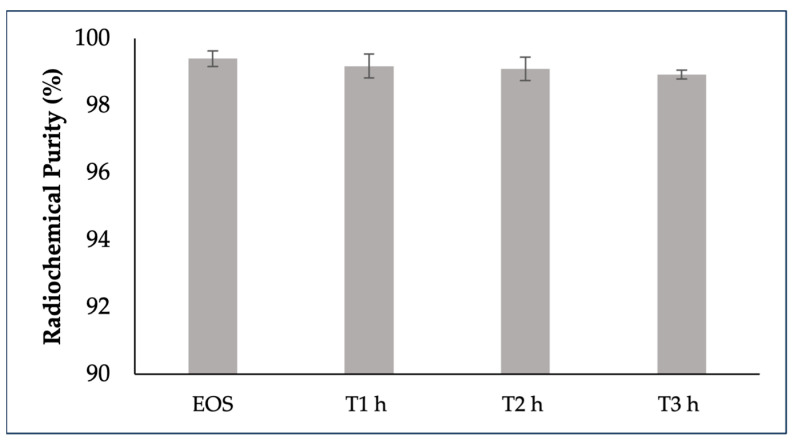
Stability of [^68^Ga]GaFAPI-46 for up to 3h after the end of synthesis. [^68^Ga]GaFAPi-46 was formulated in NaCl 0.9% with 500 mg of sodium ascorbate.

**Figure 4 ijms-24-15101-f004:**
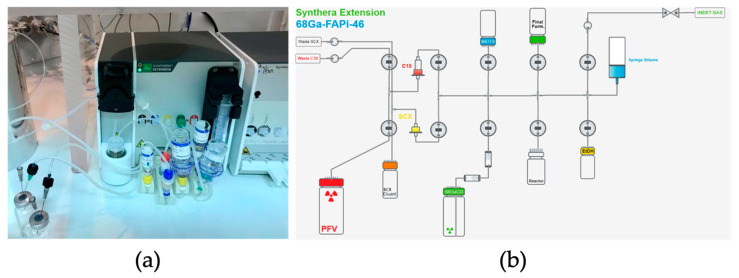
Set-up to produce [^68^Ga]GaFAPI-46 on Synthera^®^ Extension Module. (**a**) Synthera^®^ Extension module with kit and reagents installed; (**b**) layout scheme of the automatic module for the synthesis of [^68^Ga]GaFAPI-46.

**Table 1 ijms-24-15101-t001:** Results of [^68^Ga]GaFAPI-46 produced with gallium-68 from cyclotron. Total amount of peptide 50 µg, 5 min reaction time at 90 °C.

Synthesis No	Starting Activity (GBq)	Amount of Peptide (µg/GBq)	Decay Corrected RCY (%)	Purity HPLC (%)	Purity TLC (%)
1	4.68	10.67	90.20	99.61	99.86
2	4.27	11.70	91.00	98.18	99.94
3	3.96	12.62	90.04	99.04	99.13
Mean ± SD	4.31 ± 0.36	11.67 ± 0.97	90.53 ± 0.42	98.94 ± 0.72	99.64 ± 0.45

**Table 2 ijms-24-15101-t002:** Comparison of reaction conditions, decay-corrected RCY, and RCP yielded from Spreckelmeyer and Alfeimi. Conditions corresponding to the synthesis using 50 µg of peptide and gallium-68 from generator.

	Spreckelmeyer et al. [7], on MLPT	Spreckelmeyer et al. [7], on ML Eazy	Alfeimi et al. [6], on Trasis EasyOne	Alfeimi et al. [6], on Synthra	Alfeimi et al. [6], on Scintomics
Reaction Temp.	95 °C	98 °C	90 °C	90 °C	90 °C
Reaction time	10 min	10 min	4 min	4 min	4 min
RCY (%)	95.20 ± 1.40	89.70 ± 6.70	92.45	92.32	92.86
RCP (%) HPLC	99.70		99.40	99.70	99.80
RCP (%) TLC	99.90		-	-	-

**Table 3 ijms-24-15101-t003:** Summary of validation results of HPLC, TLC, and GC methods.

Validation Results Summary of the HPLC Method				
Test Parameter	Acceptance Criteria		Results	
Repeatability	6 repetitions of [^68^Ga]GaFAPI-46	RSD ≤ 5%	4.89	Appendix A
	6 repetitions of [^nat^Ga]GaFAPI-46	RSD ≤ 5%	3.40	Appendix A
Specificity/Selectivity	Resolution between peaks:			
	[^68^Ga]GaFAPI-46	5.0 ≤ RT ≤ 6.0	5.37	Appendix A
	[^nat^Ga]GaFAPI-46	5 ≤ RT ≤ 6	5.28	Appendix A
	[^68^Ga]GaCl_3_	2 ≤ RT ≤ 2	1.33	Appendix A
	RRT ([^nat^Ga]GaFAPI-46/[^68^Ga]GaFAPI-46)	0.9 ≤ RRT ≤ 1.1	0.98	
LOQ	S/N ratio ≥ 10	≤0.5 MBq/mL	10.70	Appendix A
Linearity	MBq/mL (5 concentrations)	R^2^ ≥ 0.99	1.00	Appendix A
Range	Reported Value	0.13–99.77 MBq/mL		
**Validation Results Summary of the TLC method**				
Test Parameter	**Acceptance criteria**		**Results**	
Repeatability	6 repetitions of [^nat^Ga]GaFAPI-46	RSD ≤ 0.2%	0.12	Appendix A
Specificity/Selectivity	Resolution between peaks:			
	[^68^Ga]GaFAPI-46	R/F > 0.55	0.63	Appendix A
	[^68^Ga]GaCl_3_	R/F < 0.15	0.12	Appendix A
LOQ	S/N ratio ≥ 10	(0.17 MBq/mL)	23.1	Appendix A
Linearity	MBq/mL (5 concentrations)	R^2^ ≥ 0.99	1.00	Appendix A
Range	Reported Value	0.15–47.00 MBq/mL		
**Validation Results Summary of the GC method**				
**Test Parameter**	**Acceptance criteria**		**Results**	
Repeatability	6 repetitions of [^68^Ga]GaFAPI-46	RSD ≤ 5%	1.90	Appendix A
LOQ	S/N ratio ≥ 10	(50 mg/10 mL)	247.70	Appendix A
Linearity	50–2500 mg/10 mL (6 concentrations)	R^2^ ≥ 0.99	1.00	Appendix A
Range	Reported Value	50–2500 mg/10 mL		
Precision	6 repetitions of EtOH 2500 mg/10 mL	RSD ≤ 5%	3.88	
Accuracy	Spiked conc. 1500 mg/10 mL	≤10%	6.83	

**Table 4 ijms-24-15101-t004:** Summary of the product specifications for [^68^Ga]GaFAPI-46, and quality control test of three batches for process validation.

Tests	Method	Specifications	Results (*n* = 3)
Appearance	Visual Inspection	Clear, colorless, or slightly yellow solution	Comply
pH	Potentiometric or strips	4 to 8	6.5
Identification			
Radionuclidic Identification—Energy photons γ	Gamma-ray spectrometry	The principal gamma photons have energies of 0.511 MeV and 1.077 MeV, and a sum peak of 1.022 MeV may be observed; peaks due to gamma photons with energy of 1.883 MeV may be observed.	Comply
Half-life	Ionization Chamber	61 min to 75 min	67.6
Chemical Purity			
HEPES	TLC	≤0.5 mg/10 mL	Comply
Radiochemical Purity			
[^68^Ga]GaFAPI-46	Radio-HPLC	≥95%	97.8
Peak area of gallium-68 species RF < 0.2	TLC (Radioactivity detector)	≤3%	1.3
Radionuclidic Purity			
Gallium-68	Gamma-ray spectrometry	≥98%	99.8
Gallium-66 and Gallium-67 ^1,2^	Gamma-ray spectrometry	≤2%	0.2
Other gamma-ray-emitting impurities ^1,3^	Gamma-ray spectrometry	≤0.1%	0.0
Residual Solvents			
Ethanol ^4^	GC-FID	≤2500 mg/10 mL	732.1
Biological Tests			
Endotoxin analysis	Direct inoculation	No evidence of growth should be found	Comply

^1^ According to Ph. Eur., these tests are carried out after batch release for use. ^2^ We retained the preparation to be examined for at least 12 h to allow the gallium-68 to decay to a level that would permit the detection of gallium-66 and gallium-67. ^3^ We retained the preparation to be examined for at least 24 h to allow the gallium-68 to decay to a level that would permit the detection of impurities. The peaks were disregarded due to the decay of gallium-66 and gallium-67. ^4^ According to Ph. Eur., there is a maximum of 2.5 g per administration assuming the density is 0.790 g/mL.

**Table 5 ijms-24-15101-t005:** HPLC method used to identify and determine radiochemical impurities in the [^68^Ga]GaFAPI-46 drug product. Specifications of the HPLC equipment used, model Agilent 1260 Infinity II and operation conditions of the method.

Column:	Avantor/ACE, ACE 3 C18, 3 µm, 150 × 3 mm S/N: A210625054		
Detector:	VWD 1260 Infinity II G7114A		
Data acquisition software:	Software Gina X		
Wavelength:	264 nm		
Scintillation:	Allow LOQ ≤ 0.05 MBq/mL		
Column temperature:	Room temperature (not controlled)		
Flow:	0.6 mL/min		
Injection volume:	20 µL		
Run time:	15 min		
Program:	Time (min)	% Mobile phase A	% Mobile phase B
	0.0	87	13
	15.0	87	13
Mobile Phase A:	Water/TFA = 1000/1 (*v*/*v*)		
Mobile Phase B:	Acetonitrile/TFA = 1000/1 (*v*/*v*)		
Diluent:	Water for injection		

**Table 6 ijms-24-15101-t006:** TLC method used to determine radiochemical purity of [^68^Ga]GaFAPI-46 drug product. Specifications of the TLC equipment used, model miniGita and operation conditions of the method.

Detector:		Scintillation
Data acquisition software:		TLC Control software, version 2.30
Detector	Scintillation:	Allow LOQ ≤ 0.5 MBq/mL
Others	Chromatographic paper:	Agilent iTLC-SG
	Application volume:	5 µL
	Elution length:	80 mm (origin: 1.0 cm from the bottom end; elution front: 2.0 cm from the top end)
	Mobile Phase A:	Ammonium acetate 1.0 M/methanol = 1/1 (*v*/*v*)
	Diluent:	Water for injection

**Table 7 ijms-24-15101-t007:** GC method used to quantify the presence of EtOH in the final drug product. Specifications of the GC equipment used, model 6850A, and operation conditions of the method.

Injector	
Mode	Split
Temperature	250 °C
Split ratio	15:1
Gas	Helium
Liner	Cone liner with glass wool, 4.0 mm ID, PN#5183-4647.
Oven	
Equilibrium time	0.00 min
Run time	15.0 min
Detector	
Temperature	260 °C
Mode	Constant Makeup
Makeup flow	30 mL/min (He)
Hydrogen flow	30 mL/min
Air flow	300 mL/min
Column (HP-Fast Residual Solvent; PN#1095V-420E or equivalent)	
Mode	Constant flow
Flow	3.0 mL/min
Length	30 m
Internal diameter	530 µm
Film Thickness	1.0 µm

## Data Availability

Data are contained within the article and supplementary materials.

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
