# Peer review of "Fully Automated Production of [68Ga]GaFAPI-46 with Gallium-68 from Cyclotron Using Liquid Targets"

_ijms, 2023, doi:10.3390/ijms242015101_

Round 1

Reviewer 1 Report

Compared to 68Ga/68Ge generator production, medical cyclotron produced 68GaCl3 could potentially overcome the limitation of total activity and reduce the cost. Herein, the authors developed a fully automated synthetic method of 68Ga-FAPI-46 from medical cyclotron produced 68GaCl3. 68Ga-FAPI-46 was synthesized consistently with high radiolabeling yield, radio purity and stability comparable to 68Ga/68Ge generator-based synthesis. This result could stimulate and facilitate medical cyclotron production of 68GaCl3 and clinical application of 68Ga-FAPI-46 and other 68Ga-labeled radiopharmaceuticals. I recommend this manuscript to be published on International Journal of Molecular Sciences. Please find attachment for comments and answer the following questions.

1. Please state the radiochemical yield (RCY) as decay corrected radiochemical yield.

2.  In the supplementary file, Figure S1. Representative HPLC chromatograms of each solution: (a) [68Ga]GaCl3, please explain why there is a radio peak at around retention time 5.5 min.

Reviewer 2 Report

The authors report GMP synthesis of the [68Ga]Ga-FAPI-46 using [68Ga] produced from medical cyclotron using liquid targets.. The activity, reproducibility and purity were assessed by using HPLC, TLC and GC. Authors report good repeatability and linearity of the methods used. The synthesis methods and further validation are well detailed out. The results are clearly represented and compared to available standard. The authors were able to achieve radiochemical yield similar to previous reported studies while increasing the activity around 3-fold. The high demand of [68Ga] based tracers for medical diagnosis can surely benefit from this process.

However, the major distinction of the present study, preparation of [68Ga] from liquid target using cyclotron doesn’t seem convincing novelty for publication in IJMS. This study is more of incremental improvement of already known synthesis process. Hence, I would not recommend publication of the article in IJMS.

Minor edits are needed to improve the English language of the article. 

Reviewer 3 Report

Title: Clear, concise and forceful

Abstract: A good contextualization is developed, the objective of the research work is clear, a clear idea of the methodology is presented, some important results are reported and it is concluded with a short conclusion. Describe the meaning of the following acronyms PET and GMP  

Keywords: The journal allows a greater number of words. It is advisable to include other keywords to improve the visualization of your manuscript in the future.

Introduction: Despite being brief, the authors manage to contextualize the reader, present the knowledge gap, justify the document and invite the reader to continue reading. (OK) 

 Materials and Methods:

1. Some methods are described in other articles, however it is important that the document presented has all the information, this information can be presented as complementary data. The need to cite previous works is understood when processes are carried out with the same methodology, however it is important to present the reader with complete information.

Results and Discussion:  It is necessary to report the complete validation data as supplementary material. The above, because the authors indicate in the results section that the validation of a synthesis method and the analytical methods of quality control.

Conclusion: The propositions are well argued, and supported by the results obtained.

In general terms, the work is well developed; after consulting the methodology in the referenced documents, it can be established that the quality of the results is guaranteed. However, it is necessary to report complete information on both methodology and validations, this further confirms the quality of the document.

Round 2

Reviewer 2 Report

The authors have responded satisfactorily to comments of all the reviewers. Hence, the article may be accepted for publication in IJMS. 

The quality of English language is acceptable.